# Lattice Structure Design Method Aimed at Energy Absorption Performance Based on Bionic Design

**Gang He \***⬥**, Hu Yang, Tao Chen, Yuan Ning, Huatao Zou and Feng Zhu**

Department of Mechanics, College of Mechanical & Electrical Engineering, Hohai University,
Changzhou 213022, China
* Correspondence: hegang@hhu.edu.cn

**Abstract:** To obtain the lattice structure with excellent energy absorption performance, the structure of loofah inner fiber is studied to develop bionic design of lattice structure by experiment and simulation analysis method. From the compression experiment about the four bionic multi-cell lattice structures (bio-45, bio-60, bio-75, and bio-90) and VC lattice structures, we found that all are made of PLA and fabricated by the fused deposition modeling (FDM) 3D printer. The comprehensive performance of bio-90 lattice structure is the best in the performance of the specific volume energy absorption ($SEA_v$), the effective energy absorption ($EA$), and the specific energy absorption ($SEA$). Based on the experimental result, the energy absorption performance of bio-90 lattice structure is then studied by the simulation analysis of influence on multiple parameters, such as the number of cells, the relative density, the impact velocity, and the material. The results can provide a reference for the design of highly efficient energy absorption structures.

**Keywords:** lattice structure; loofah; energy absorption performance; bionic design; relative density





## 1. Introduction

Lattice structure is the space truss structure, which belongs to the scope of porous structure, and is composed of nodes, rods, and panels on a certain rule. It was first proposed by Ashby [1] of Cambridge University and Evans [2] of Harvard University in 2001. Lattice structures have gained extensive attention for their comprehensive properties such as their light weight, high strength, and impact resistance. Their open internal spaces make them have the advantage in load bearing and energy absorption, and also exhibit better properties than foam structures [3]. Several typical lattice structures [4] are shown as Figure 1. Cao [5] modified the same diameter core rod of single-cell structure into a variable cross-section core rod and proposed an improved single-cell structure. Bai [6] used PA2200 as a raw material, printed three experimental samples of body-centered cubic (BCC), rod-diameter-change graded body-centered cubic (RGBCC) and size-change graded body-centered cubic (SGBCC), and studied the effect of gradient direction on the structure. Andrew [7] discussed the influence of impact energy, relative density, plate thickness, and impact angle on the dynamic impact behavior of the lattice structure by the weight drop experiment. Hammetter [8] studied the effects of slenderness ratio, inclination angle, and single-cell layer on the compression deformation mechanism of the lattice structure.

The lattice structures are considered as the most promising structure–function integrated structures at present. Recently, the development of additive manufacturing technology has resulted in the possibility of the manufacture of a complex inner structure, and that makes it is flexible to product design. To obtain special performance [9,10], the method of design about lattice structures has become a hot research topic. Squid bones, beetle elytra, glass sponges, and bamboos are found to have porous features at the microscopic level and exhibit excellent comprehensive performance, which gives researchers substantial inspiration [11,12]. After extensive studies, the biomimetic method has been proven to be an effective way to design lattice structures with better properties.

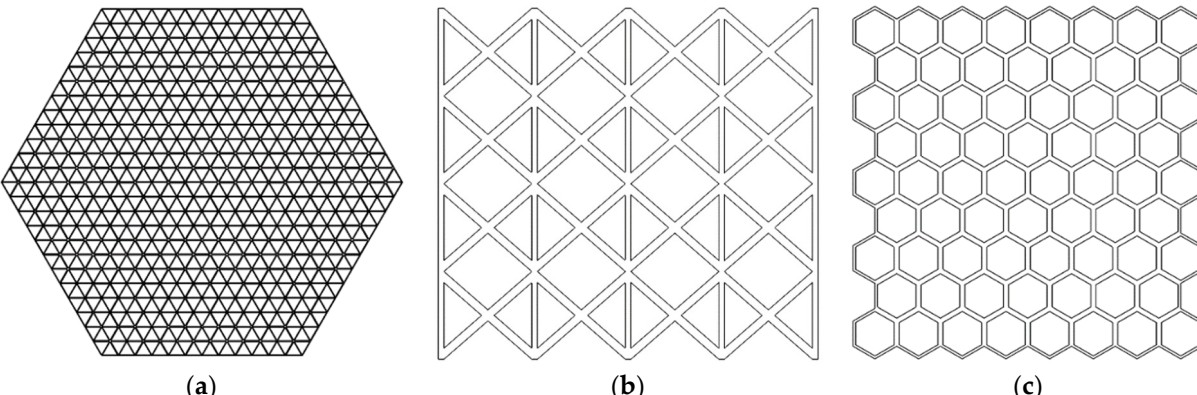

**Figure 1.** Several typical lattice structures: (**a**) SC lattice structure; (**b**) FC lattice structure; (**c**) Honeycomb structure.

Based on the excellent performance of the lattice structure, it is widely used in aerospace, safety protection, medical, and other fields. In the aerospace field, it has always been a major design goal that the structure is lightweight and has smaller material consumption, less fuel consumption, and higher performance. The excellent performance of lattice structure means it has great application value in the aerospace field. Helmet lining is the main component of the helmet; it can dissipate impact energy and reduce the load transmitted to the head in case of accident. SOE and HEXR [13] proposed a honeycomb-lined helmet and found that the impact energy was better distributed and absorbed by the honeycomb-lined helmet, and it had great potential in improving the safety of the helmet. The KOLLODE alliance and four innovative companies (KUPOL, TAXIX, ShapeShift3d, Numalogics) used virtual design and 3D printing technology to manufacture a helmet liner KUPOL, which can absorb energy and redirect impact force.

Loofah is highly porous material, and it has great application potential in sound absorption, shock absorption, and cushioning. Due to the unique fiber structure, loofah can bear a large load, and Zou [14] made an optimization algorithm on the displacement and stress relationship of the geometric structure. For studying the mechanical properties of loofah, Wang [15] found that the inner surface of the loofah plays a major role in supporting the axial load by conducting the tests of quasi-static compression and dynamic impact on the loofah structure. Elmadih [16] researched the ability of these lattices to provide vibration attenuation at frequencies greater than their natural frequency. Chen [17] et al. performed a multi-scale study to explore the relationship between the structure and mechanical properties of different layers of fibers (inner fibers and outer fibers) and different directions (transverse and longitudinal), and the results showed that the inner ring wall fibers contribute the most to the longitudinal properties of the loofah, while the mechanical strength of the core fiber is lower than that of the inner fiber. Qing [18] designed a new type of ultra-light bionic tube structure by combining the structure of the loofah and the pores of honeycomb hexagons, and found that its equivalent elastic modulus is from 166.9 to 180.59 MPa by the compression test.

At present, most bionic designs are aimed at the macrostructure of loofah. The fiber structure of loofah is the main factor affecting its mechanical properties. In this paper, the bionic design method is used to design the single-cell lattice structure. It is studied the influence of different parameters on the energy absorption of the lattice structure, such as the number of cells, the relative density, the impact velocity, and material. The results are useful for the design of a highly efficient energy absorption structure.

## 2. Modeling of Single-Cell Structure Loofah Lattice

The loofah is a natural fiber network structure [19], and it is mainly composed of the outer layer, the middle layer, the inner layer, and the core layer, as shown in Figure 2.

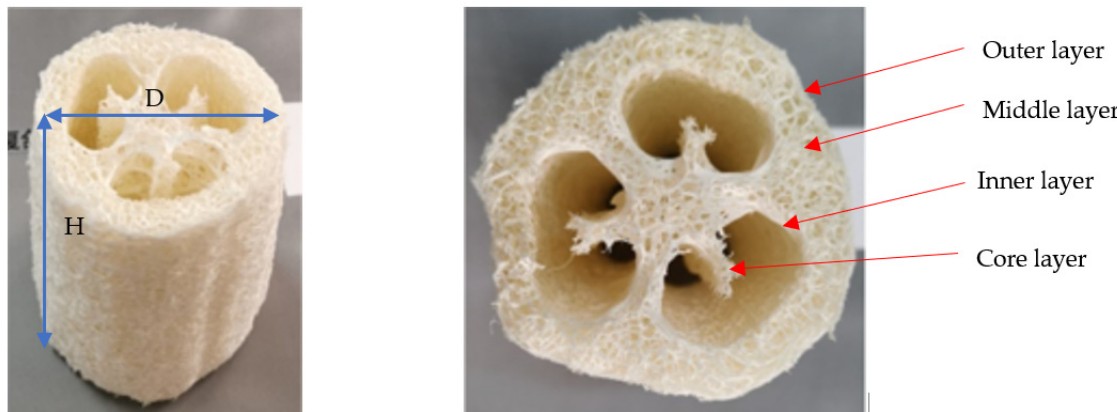

**Figure 2.** Overall structure of loofah.

Although the fiber structure of loofah is interlaced, its distribution is not disordered. The growth direction of fiber bundles in different layers is different. Most of the outer layer fibers of the loofah structure grow along the circumferential direction in a circular shape; the inner layer fibers mainly distribute along the longitudinal axis of the loofah structure and have the large diameter and regular texture; the fibers of the core layer are interwoven, forming a honeycomb structure, as shown in Figure 3.

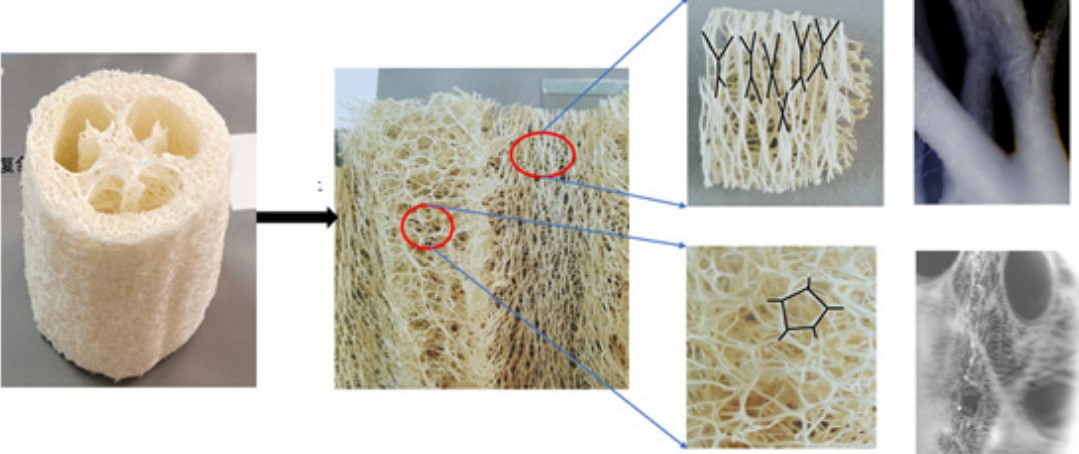

**Figure 3.** Regional structure of loofah sponge.

We extracted and simplified the textures of loofah fiber to obtain its structural features and summarized them into the 2D configurations of type I and type II, as Figure 4a shown. The inner layer fibers are actually interconnected in 3D space. The simplified spatial structure is shown as Figure 4b. After giving a certain rod diameter, the 3D single-cell structure is shown as Figure 4c.

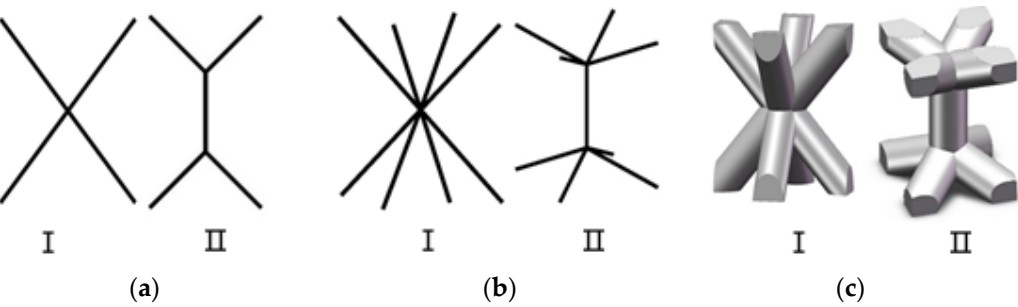

**Figure 4.** Evolution of bionic single-cell structure: (**a**) 2D structure; (**b**) 3D structure; (**c**) cell structure.

The difference between type I and type II structure is the values of angle. The type I represents the angle of 45° lattice structure, and given that the three lattice structures with angles of 60°, 75°, and 90° have uniform characteristics, they are represented by type II. By setting angles of 45°, 60°, 75°, and 90°, the series of the single-cell structure is built, and they are respectively named as bio-45, bio-60, bio-75, and bio-90, as shown in Table 1.

**Table 1.** Bionic cells with different configurations.

| Number | Angle | 2D Structure | 3D Structure |
|--------|-------|--------------|--------------|
| 1 | 45° |  |  |
| 2 | 60° |  |  |
| 3 | 75° |  |  |
| 4 | 90° |  |  |

The mechanical properties of lattice structures are mainly determined by their material and structural, and the main parameter of structural is the relative density. The relative density is the ratio of the solid volume in a single-cell structure to the volume of the cube, and its calculation formula is shown as Equation (1):

$$\overline{\rho} = \frac{V_s}{V} \times 100\% \tag{1}$$

where $\overline{\rho}$ denotes relative density of the single-cell structure, and $V_s$ and $V$ respectively denote the solid volume of the single-cell structure and the total volume of the cube.

The relative density is adjustable for a specific cell structure. In this section, the relative density of single-cell structure is theoretically deduced, and the specific derivation process of the bio-45 single-cell structure is given. There are two variable structural parameters in the bio-45 single-cell structure, which are the cell size and the core rod diameter. From Figure 5, the cell structure has eight of the same core rods.

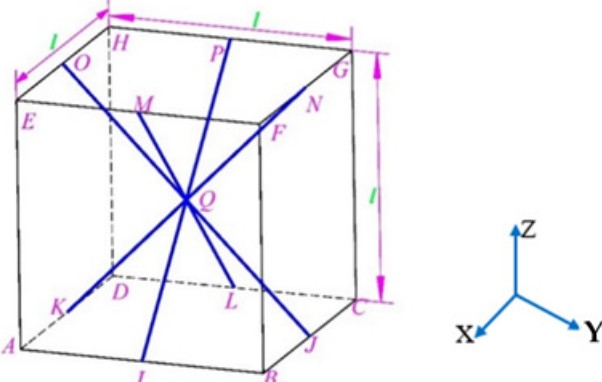

**Figure 5.** Lattice cell diagram.

According to the geometric relationship, the length of each rod of the bio-45 single-cell structure can be calculated as Equation (2):

$$l_0 = \frac{\sqrt{l^2 + l^2}}{2} = \frac{\sqrt{2}}{2}l \tag{2}$$

where $l_0$ denotes the length of each rod of the bio-45 single-cell structure, and $l$ denotes the length, width, and height of the cell structure.

The solid volume of the single-cell structure is given as Equation (3):

$$Vs = 8\pi \frac{d_0^2}{4}l_0 = \sqrt{2}\pi d_0^2 l \tag{3}$$

where $d_0$ denotes the diameter of the core rod.

The spatial volume of the single-cell structure is shown as Equation (4):

$$V_{bio-45} = l^3 \tag{4}$$

Based on the Equations (3) and (4), the relative density of the single-cell structure is as follows:

$$\bar{\rho} = \frac{Vs}{V_{bio-45}} = \frac{\sqrt{2}\pi d_0^2}{l^2}$$

Based on theoretical derivation, the calculation formula of relative density about each of the four kinds of single-cell structure can be obtained in Table 2.

**Table 2.** Calculation formula of relative density of the single-cell structure.

| Single-Cell Type | Relative Density |
|---|---|
| bio-45 | $\bar{\rho} = \sqrt{2}\pi d_0^2 / l^2$ |
| bio-60 | $\bar{\rho} = \left(7\sqrt{3} + 3\right)\pi d_0^2 / 12l^2$ |
| bio-75 | $\bar{\rho} = 1.219\pi d_0^2 / l^2$ |
| bio-90 | $\bar{\rho} = 3\pi d_0^2 / 4l^2$ |

Without considering the overlapping part of the connection between the rods and the volume at the end of rods, as shown in Figure 6, with the different edge lengths of the cube, there are differences between the calculation formula of relative density and actual relative density based on the actual relative density obtained by the volume evaluation module of the modeling software. The calculation formula of the relative density of the single-cell structures is modified by the theoretical derivation and polynomial fitting method, as shown in Table 3. The deviation between the relative density calculated by the two

formulas and the actual density is shown in Figure 7. The error of the result calculated by the modified formula is within 0.08%.

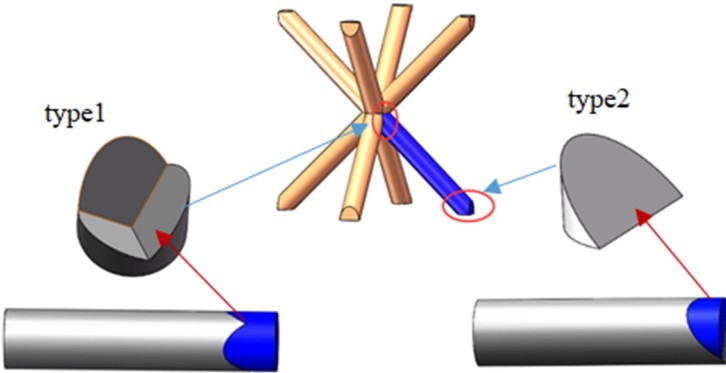

**Figure 6.** Volume to be subtracted in correction.

**Table 3.** Modified calculation formula of relative density.

| Single-Cell Type | Relative Density |
|---|---|
| bio-45 | $\bar{\rho}' = \bar{\rho} - \left[ \left( \begin{array}{c} -3.564\mathrm{e}^{-3} + 1.1264\mathrm{e}^{-3}d_0 \\ +9.72\mathrm{e}^{-4}d_0^2 + 3.909d_0{}^3 \end{array} \right) / l^3 \right]$ |
| bio-60 | $\bar{\rho}' = \bar{\rho} - \left[ \left( \begin{array}{c} 1.295 - 2.505d_0 \\ +1.39d_0^2 + 3.449d_0{}^3 \end{array} \right) / l^3 \right]$ |
| bio-75 | $\bar{\rho}' = \bar{\rho} - \left[ \left( \begin{array}{c} 18.81 - 33.2d_0 \\ +18.22d_0^2 + 1.551d_0{}^3 \end{array} \right) / l^3 \right]$ |
| bio-90 | $\bar{\rho}' = \bar{\rho} - \left[ \left( \begin{array}{c} -5.968\mathrm{e}^{-4} - 7.528\mathrm{e}^{-4}d_0 \\ -1.159\mathrm{e}^{-2}d_0^2 + 1.414d_0{}^3 \end{array} \right) / l^3 \right]$ |

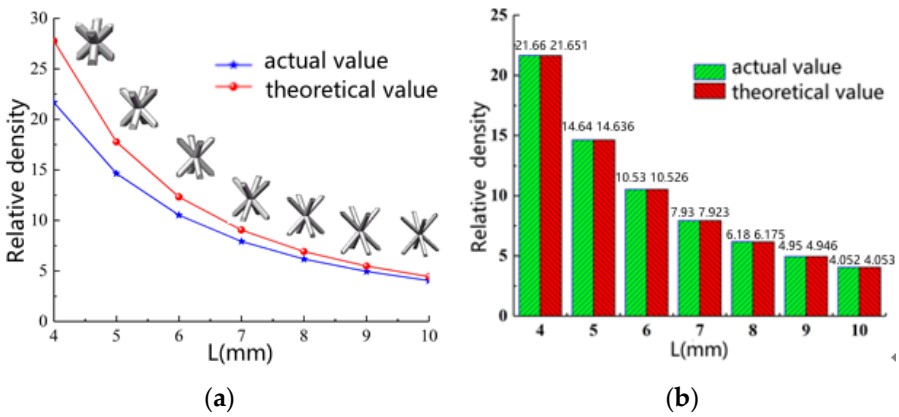

| (**a**) | (**b**) |
|---|---|

**Figure 7.** The differences between the theoretical and actual relative density: (**a**) primary formula; (**b**) modified formula.

## 3. Experiment of Static Energy Absorption Characteristics about Loofah Lattice Structure

The above four types of single-cell structure and the Vertex Cube (VC) obtained by topology optimization [20] are used to design multi-cell lattice structures, which are shown in Figure 8a, and their parameters are given in Table 4. The samples are arranged using a Solidworks software [21] and made by a fused deposition molding (FDM) 3D printer, as shown in Figure 8b. The 3D printer is the A8 equipment of the JGMaker company, as shown in Figure 9. The precision of 3D printer is 0.05~0.2mm. In order to reduce manufacturing deviation, the printing temperature, speed, printing thickness, and other parameters are all

in accordance with the recommended values of the equipment. The printing direction of the printer is bottom-up, and the different printing directions of the adjacent two layers lead to structural anisotropy. The five lattice structures used in this paper all use PLA as the support materials.

**Table 4.** Lattice structure parameters.

| Name of Lattice Structure | Length × Width × Height (mm) | Core Rod Diameter | Relative Density |
|---|---|---|---|
| bio-45 | 20 × 20 × 20 | 3.036 | 0.3 |
| bio-60 | 20 × 20 × 20 | 3.305 | 0.3 |
| bio-75 | 20 × 20 × 20 | 3.702 | 0.3 |
| bio-90 | 20 × 20 × 20 | 4.112 | 0.3 |
| VC | 20 × 20 × 20 | 4.145 | 0.3 |

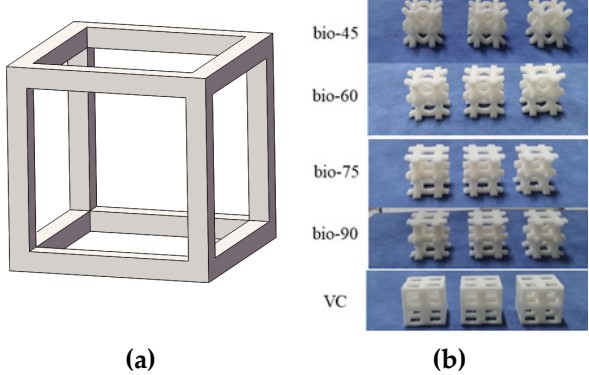

**(a)**          **(b)**

**Figure 8.** (**a**) 3D structure of Vertex Cube (VC); (**b**) samples of five different lattice structures.

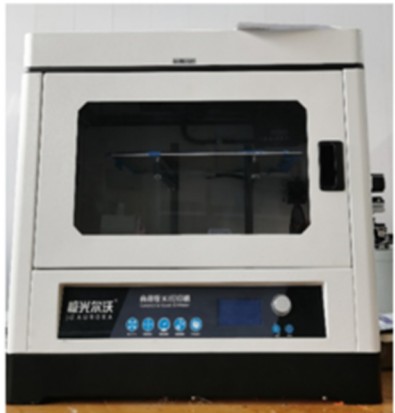

**Figure 9.** Molding equipment.

By the DZ-101 machine of Dazhong Instrument Co., Ltd., this quasi-static compression experiment of the above five models (bio-45, bio-60, bio-75,bio-90, and VC) is conducted, as shown in Figure 10, and the test is divided into three steps:

(1) Put the five models on the pressure table of DZ-101 machine, and adjust the position of the compression head to make it in good contact with the model.

(2) Start the machine, and set the compression speed of the indenter as 2 mm/min; obtain the curve from the testing machine during the compression process.

(3) After the model is compacted, replace the sample and continue the above steps until all samples are tested. Stop the compression experiment and close the machine.

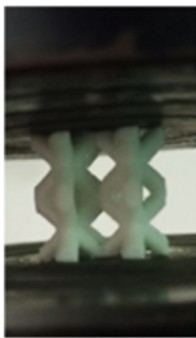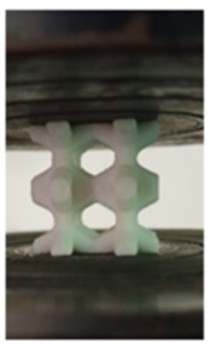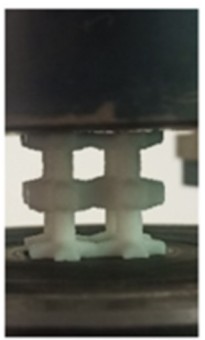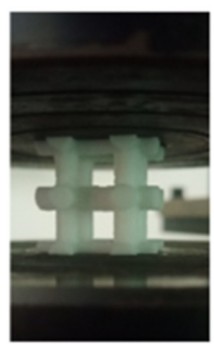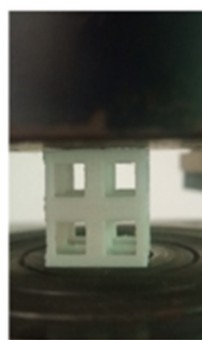

**Figure 10.** Compression test specimen clamping.

The quasi-static compression experiment can measure the mechanical response and the energy absorption characteristics of the lattice structure specimen under static load. The displacement of the compression head and the counterforce are recorded in the tests, and the stress–strain curves about five different lattice structures of bio-45, bio-60, bio-75, bio-90, and VC are shown as Figure 11. The curves are overall continuity without obvious fluctuations, which indicate that the mainly failure forms of the lattice structures are plastic deformation failures and no serious brittle fracture.

As Figure 11 shown, the compression process has gone through three stages: the elastic stage, the platform stage, and the densification stage. In the densification stage, the load changes suddenly, and the lattice structure has completely failed. During this stage, the absorbed energy has no meaning, and densification is generally regarded as the end point of energy absorption [22]. Based on the elastic phase data of the five structures, their elastic modulus and the yield strength modulus ratio can be obtained as shown in Table 5.

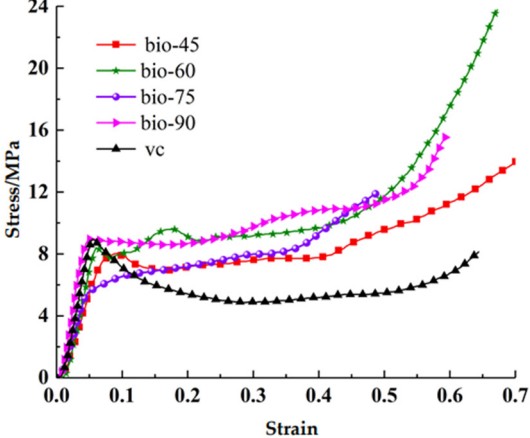

**Figure 11.** Testing stress–strain curves.

**Table 5.** Test results.

|  | bio-45 | bio-60 | bio-75 | bio-90 | VC |
|---|---|---|---|---|---|
| Elastic modulus (MPa) | 159.19 | 184.11 | 154.73 | 204.92 | 198.73 |
| Yield strength (MPa) | 8.076 | 8.379 | 5.590 | 9.028 | 8.999 |

The elastic modulus and yield strength have no relationship with the design angle, as Table 5 shown. The elastic modulus of the bio-90 lattice structure is 32.44% higher than that of the bio-75, and only 3.02% higher than that of VC lattice structure. The yield strength of the bio-90 lattice structure is 61.50% higher than that of the bio-75 structure, 11.79% higher than that of the bio-45 structure, 7.75% higher than that of the bio-60 structure, and 0.32%

higher than that of the VC lattice structure. The initial stiffness and strength of the bio-90 lattice structure are the highest among the five structures.

For evaluating the energy absorption performance of lattice structure, it is necessary to define several indexes [23], such as specific volume energy absorption, effective energy absorption, and specific energy absorption. The definition of specific volume energy absorption is

$$SEAv = \int_0^{\varepsilon d} \sigma d\varepsilon \tag{5}$$

where $SEAv$ denotes specific volume energy absorption, $\sigma$ denotes structural stress, and $\varepsilon d$ denotes densification strain. The definition of effective energy absorption is

$$EA = (SEAv) \times V \tag{6}$$

The definition of specific energy absorption is

$$SEA = \frac{(EA)}{m} \tag{7}$$

where $m$ denotes mass. The energy absorption efficiency can be defined as

$$\eta(\varepsilon) = \frac{1}{\sigma(\varepsilon)} \int_0^{\varepsilon d} \sigma(\varepsilon) d\varepsilon = \frac{(SEAv)}{\sigma(\varepsilon)} \tag{8}$$

where $\sigma(\varepsilon)$ denotes load. The platforms stress can be defined as

$$\sigma_p = \frac{1}{\varepsilon_d} \int_0^{\varepsilon_d} \sigma(\varepsilon) d\varepsilon \tag{9}$$

The energy absorption efficiency–strain relationships [23,24] of the five structures are shown as Figure 12. The energy absorption efficiency of the bio-90 lattice structure is higher than that of the other lattice structures, and this difference becomes larger with the increase of strain. From Figure 13, the specific energy absorption (*SEA*), the specific volume energy absorption (*SEA$_v$*), and effective energy absorption (*EA)* values of the bio-90 lattice structure are greater than those of the other four lattice structures; the *SEA* value of the bio-90 lattice structure is 2.335 times greater than that of the VC lattice structure, the *SEA$_v$* value is 1.933 times, and the *EA* value is 1.933 times.

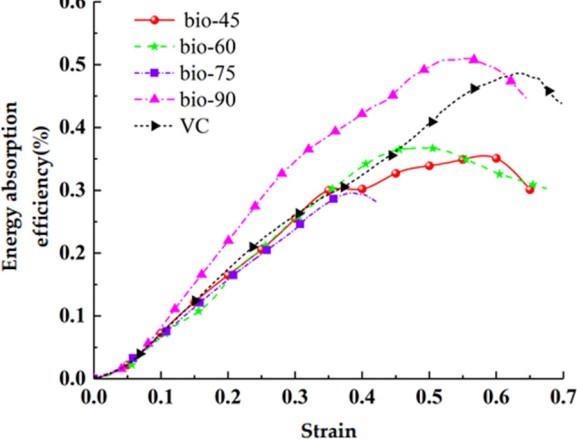

**Figure 12.** Energy absorption efficiency.

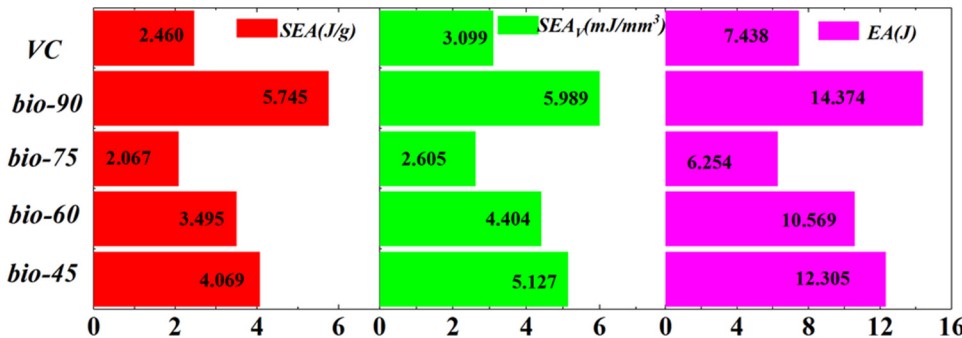

**Figure 13.** Comparison of *SEA*, *SEA_v*, and *EA*.

The *SEA*, *SEA_v*, and *EA* values of the bio-75 lattice structure are the smallest, which are respectively 35.98%, 43.50%, and 43.51% of those of the bio-90 lattice structure. In summary, the bio-90 lattice structure has the best energy absorption performance among the five lattice structures, not only in the stable plateau phase, but also in the specific energy absorption (*SEA*) and other indexes.

## 4. Analysis with the Influence of Dynamic Performance Parameters about Loofah Lattice Structure

In order to verify the accuracy of the simulation results, the quasi-static compression simulation analysis is carried out through ANSYS software, and the results are compared with the stress and strain results in Section 3. First, establish the lattice cell model in the Spaceclaim module, and obtain the overall size of 20 mm × 20 mm × 20 mm lattice structure, such as the model of bio-45 lattice structure, which is shown in Figure 14a.

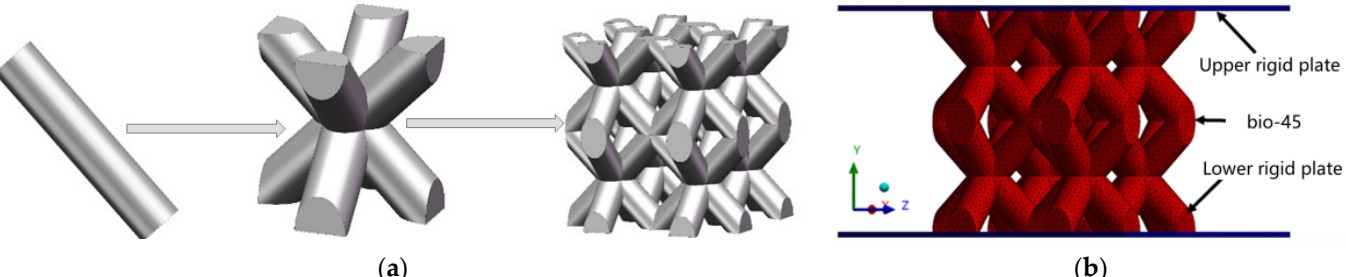

(**a**)                               (**b**)

**Figure 14.** (**a**) The lattice structure of bio-45; (**b**) the mesh of bio-45.

The material is PLA, and the lattice structure is anisotropic, which is the same as the 3D-printed lattice structure. Two steel plates are placed on the upper and lower surfaces of the lattice structure, respectively. The steel plate and the lattice structure are in friction contact, with a friction coefficient of 0.15. The constraint condition is that the lower steel plate is fixed, and the rotation of the upper steel plate in the X, Y, and Z directions is limited. The tetrahedral element grid is adopted for grid division, with the grid size of 0.6mm and the number of elements of 106325, as shown in Figure 14b. The modeling and mesh generation processes of the bio-60, bio-75, bio-90, and VC lattice structures are the same as the bio-45, and results of the finite element analysis are shown as Figure 15.

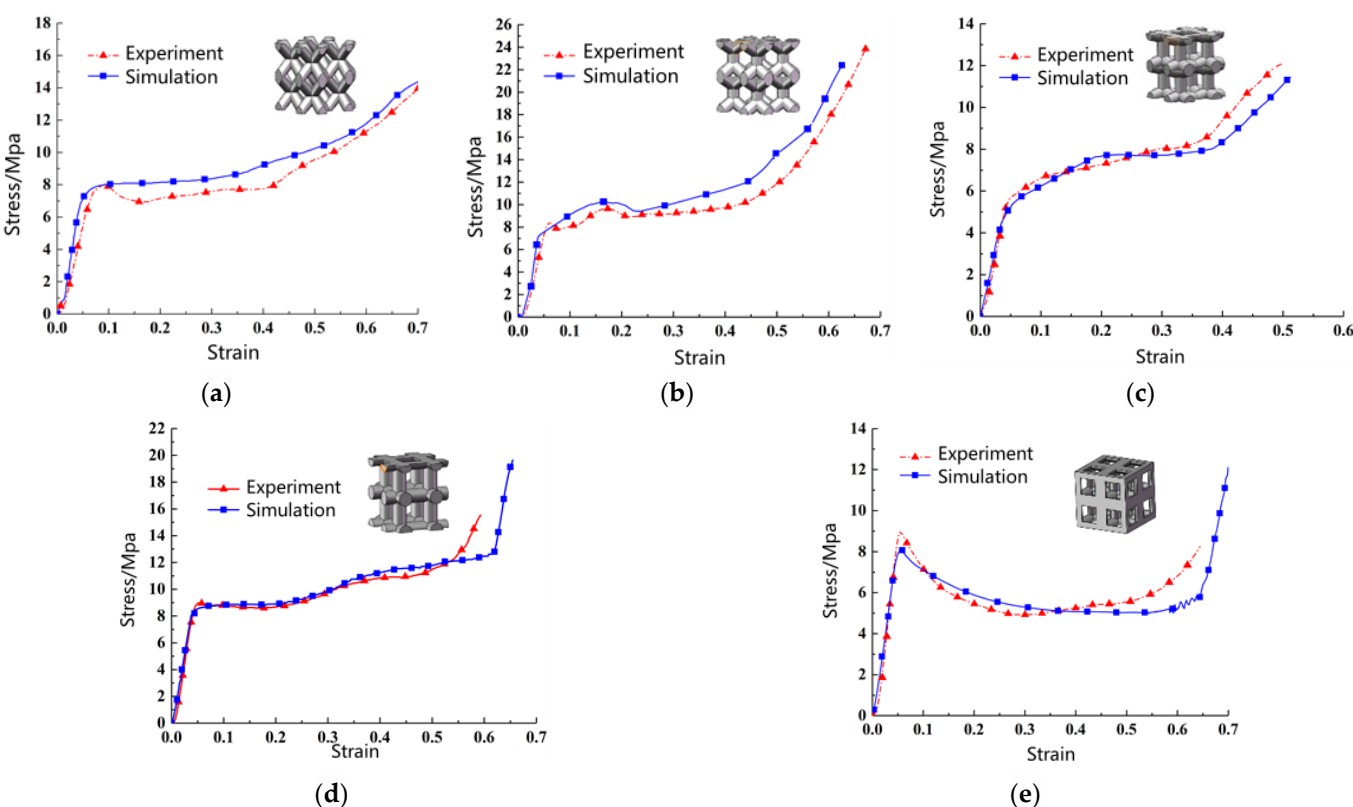

**Figure 15.** Stress–strain curve of experiment and simulation: (**a**) bio-45; (**b**) bio-60; (**c**) bio-75; (**d**) bio-90; (**e**) VC.

As Figure 15 shown, the relative errors of the simulation and test are within the acceptable range, and the maximum relative errors of the five structures are about 10%, indicating that the simulation results are similar to that of experiment, which verifies the reliability of the finite element model. Then, the influence of the number of cells, relative density, impact velocity, and material on the energy absorption characteristics of the lattice structure is analyzed with the finite element method.

### 4.1. The Influence of the Number of Cells on the Energy Absorption Characteristics

Keeping the relative density of the bio-90 bionic lattice structure at 0.3, the impact velocity is 30 m/s, and the single-cell structures are arranged by the arrays of $2 \times 2 \times 2$, $3 \times 3 \times 3$, $4 \times 4 \times 4$, $5 \times 5 \times 5$, $6 \times 6 \times 6$, and $7 \times 7 \times 7$, respectively, to get the six lattice structures of different overall sizes, the finite element model is shown as Figure 16.

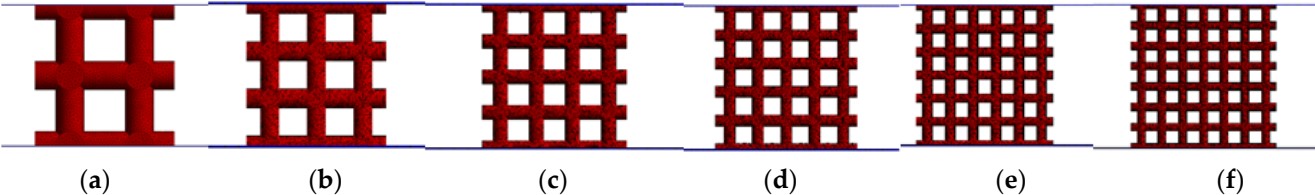

**Figure 16.** Finite element model of lattice structure: (**a**) 8-cell structure; (**b**) 27-cell structure; (**c**) 64-cell structure; (**d**) 125-cell structure; (**e**) 216-cell structure; (**f**) 343-cell structure.

The simulation analysis results of the bio-90 lattice structure with different cell numbers are shown in Figures 17 and 18. Figure 17 shows the stress–strain curves of the lattice structure with different cell numbers. They still have an obvious elastic phase, a plateau phase, and a densification phase at the impact velocity of 30 m/s, which are similar to the results of the quasi-static compression.

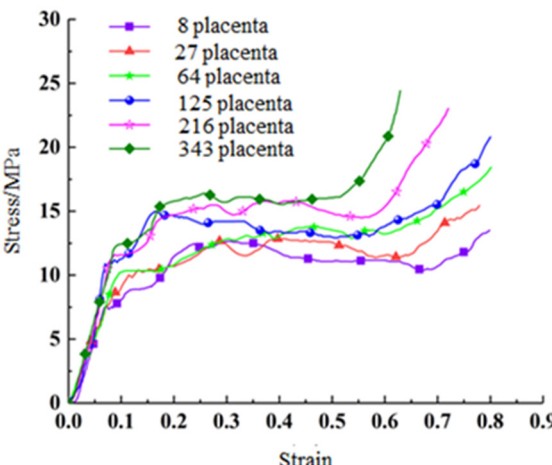

**Figure 17.** Stress–strain curves with different number of cells.

Crash load efficiency is defined as

$$CLE = \frac{(MCF)}{(MIF)} \times 100\% \tag{10}$$

where *MCF* denotes maximum current force, and *MIF* denotes maximum instantaneous force, which is defined as

$$MCF = \frac{(EA)}{S} \tag{11}$$

where *S* denotes effective displacement.

The slope of the stress–strain curves of the lattice structure with different cell numbers in the elastic phase is almost the same. This indicates that the number of single cells has little effect on the elastic phase of the structure, which is consistent with the analytical conclusion of the literature [12].

As shown in Figure 18, with the number of cell elements of the lattice structure increasing, the specific volume energy absorption ($SEA_v$), the specific energy absorption ($SEA$), the crash load efficiency ($CLE$), and platform stress ($\sigma_p$) increase and then decrease. However, in general, the number of single cells does not have a significant effect on the above four indexes.

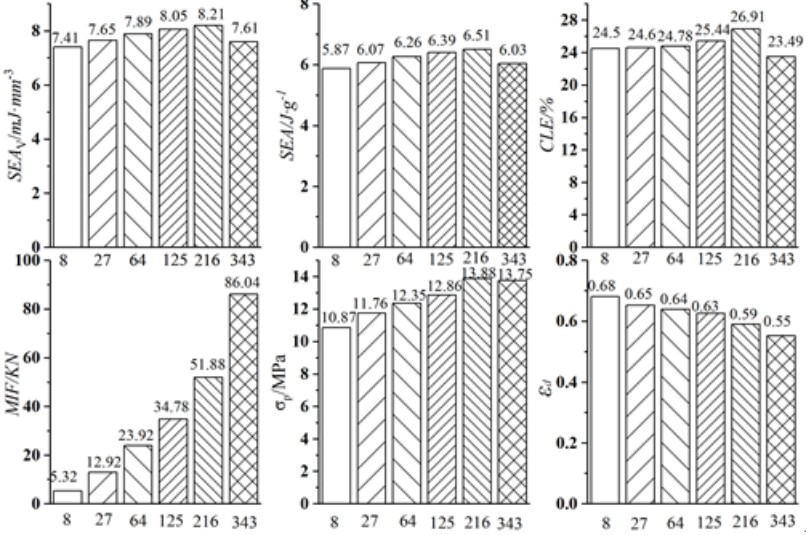

**Figure 18.** Energy absorption index with different number of cells.

### 4.2. The influence of Relative Density on Energy Absorption Characteristics

To investigate the effect of relative density on the energy absorption performance of the lattice structure, we use six relative densities of 0.09, 0.16, 0.23, 0.30, 0.37, and 0.44 with 125-grid lattice structures and the impact velocity of 30 m/s. The stress–strain curves and the indexes of $SEA_v$, $SEA$, $CLE$, $MIF$, $\sigma_p$, and $\varepsilon_d$ at different relative densities are shown in Figures 19 and 20.

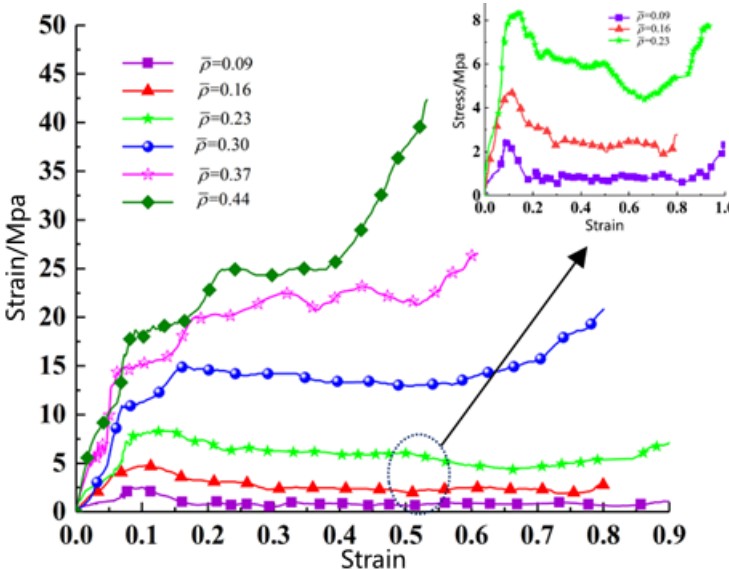

**Figure 19.** Stress–strain curves under different relative densities.

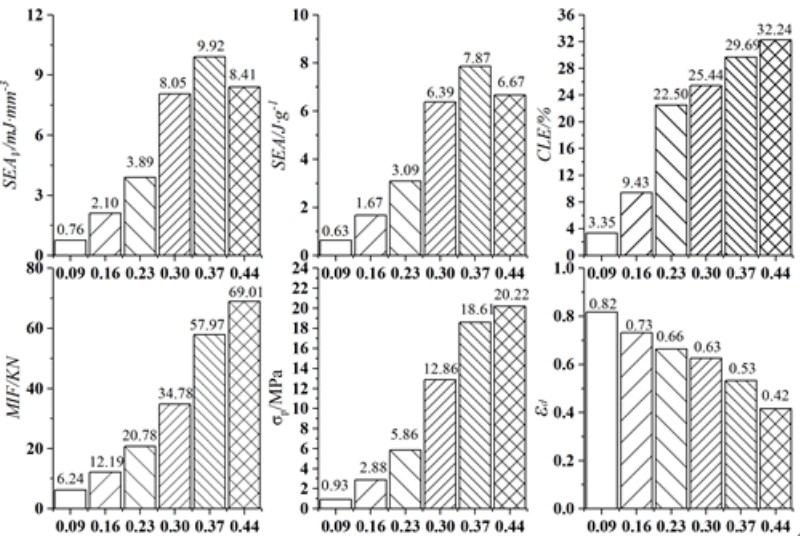

**Figure 20.** Energy absorption index at different relative densities.

As Figure 19 shown, at the elastic phase, the slope of the stress–strain curve becomes steeper with the relative density increasing, which indicates that the initial stiffness of the lattice structure is higher. At the platform phase, with the relative density increasing, the corresponding platform stress increases more obviously, and the overall stress–strain curve rises. At the densification phase, the overall stress–strain curve rises with the increase of the relative density, and the structure will reach densification in advance to a certain extent, which is negative to the energy absorption of the structure.

As Figure 20 shown, when the relative density is less than 0.37, with the relative density increasing, the specific volume energy absorption ($SEA_v$) and the specific energy absorption

(*SEA*) increase. However, when the relative density is 0.44, the specific volume energy absorption (*SEA*$_v$) and the specific energy absorption (*SEA*) are smaller than the values of those when the relative density is 0.37. This is because, although there is a relatively high-stress plateau at this point, the increase in relative density makes the strain of the structure to reach the densification stage much lower and the structure transitions to the densification stage earlier.

### 4.3. The influence of Impact Velocity on Energy Absorption Characteristics

In order to study the effect of impact velocity on the energy absorption performance of the lattice structure, the impact simulations of the lattice structure are performed under six different velocities of 10 m/s, 30 m/s, 50 m/s, 70 m/s, 90 m/s, and 110 m/s, with the relative density of 0.23 and cell elements of 125. The stress–strain curves of the lattice structure are shown as Figure 21. From Figure 21, at the six impact velocities, the overall trend of stress–strain curves of the lattice structure with specific relative density is relatively similar, the overall stress–strain curves under high speed are higher than those under low speed, and the height is consistent at the elastic phase of the compression process.

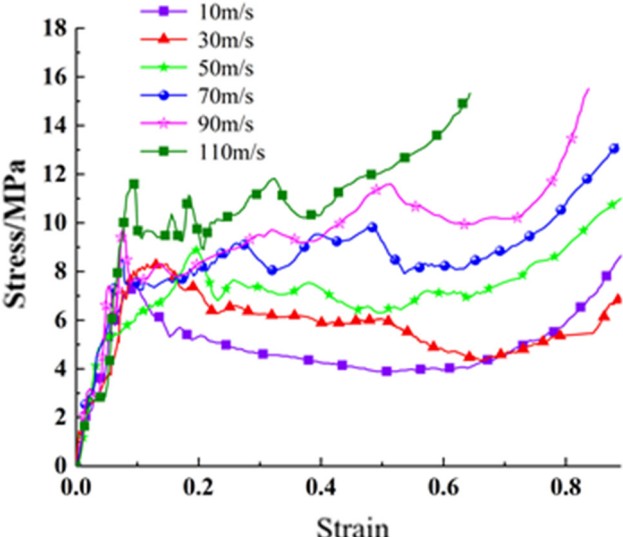

**Figure 21.** Stress–strain curves at different impact velocities.

With the impact velocity increasing, the strain rate sensitivity is higher. At the plastic stage, the difference between the curves gradually increases, the load becomes unstable, and there is greater fluctuation of its stress–strain curve. This phenomenon is particularly significant after v = 50 m/s.

By calculating various energy absorption and other indicators at six different impact speeds, the results about the discussion of the energy absorption characteristics of the lattice structure under dynamic impact response are shown in Figure 22.

With the impact velocities increasing from 10 m/s to 110 m/s, the densification strain of the structure increases from 0.64 to 0.74, while the specific volume energy absorption (*SEA*$_v$), specific energy absorption (*SEA*), and maximum instantaneous force (*MIF*) increase monotonically from 2.90 mJ·mm$^{-3}$ to 6.92 mJ·mm$^{-3}$, 2.30J·g$^{-1}$ to 5.49J·g$^{-1}$, and 18.38 kN to 29.55 kN, respectively. On the contrary, the crash load efficiency (*CLE*) always decreases. Compared with that at the velocity of 10 m/s, the crash load efficiency (*CLE*) at v = 110 m/s decreases 39.03%.The results show that the higher impact velocity, the more unstable energy absorption process of the structure, which is also consistent with the actual experiment process.

Since the different impact velocities affecting the force state of the structure at the beginning, the impact velocities will have a significant effect on the deformation mode of the dotted structure, which is the fundamental reason for changing the energy absorption

performance of the structure. The deformation mode of the structure under different impact velocities is shown in Figure 23.

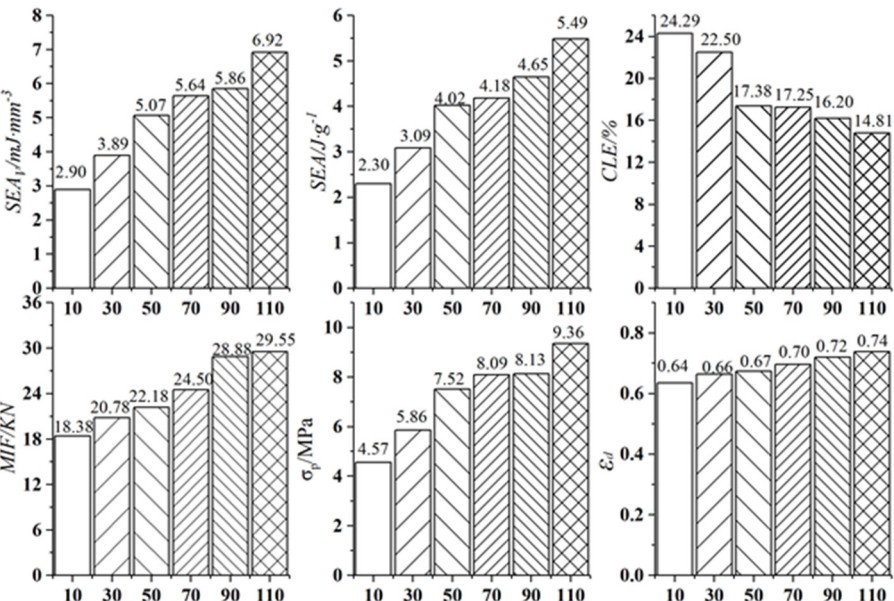

**Figure 22.** Energy absorption index under different impact velocities.

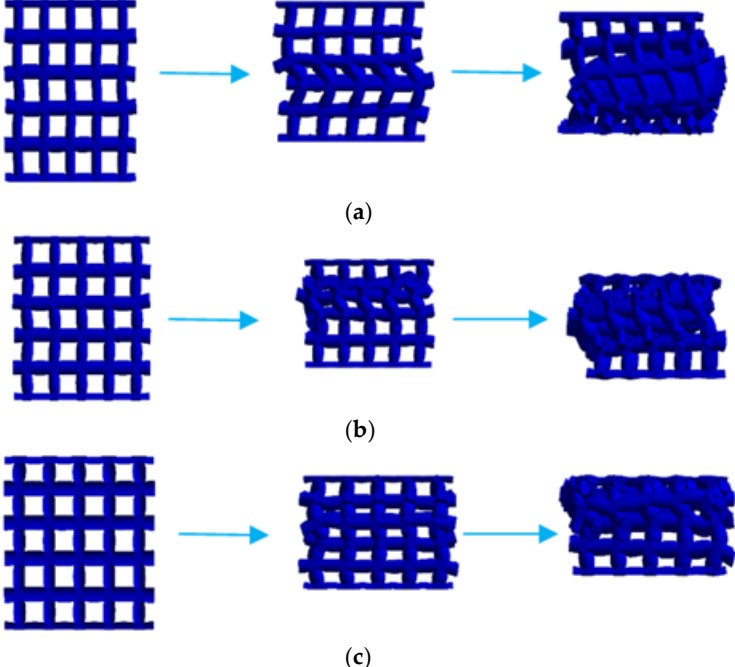

**Figure 23.** Deformation of structure under different impact velocities. (**a**) v = 10 m/s, (**b**) v = 50 m/s, and (**c**) v = 90 m/s.

As Figure 23a shown, when the impact velocity is v = 10 m/s, the cell element in the middle part of the structure first shows bending deformation. With the compression process continuing, the deformation is rapidly transferred to the fixed end, and the overall structural deformation presents the characteristic of tilting to one side until the structure compresses to densification. As seen in Figure 23b, when the impact velocity is 50 m/s, the initial deformation of the structure still occurs in the middle and lower part. With the compression continuing, the deformation gradually moves to the impact end, and then

the impact end is first compacted. When the impact velocity is 90 m/s, the impact end is first compacted, and then the deformation gradually passes to the fixed end. The result is shown in Figure 23c.

### 4.4. The influence of Material on Energy Absorption Characteristics

The parameters of the base material have an important influence on the performance of the lattice structure. The influence of the base material on the dynamic impact performance of the lattice structure is studied by investigating the specific response of the lattice structure with four different base materials under dynamic impact. The parameters of four materials are listed in Table 6. The dimensions of the analyzed models are the same: all are 125-cell structures with the relative density of 0.23 and the impact velocity of 30 m/s.

**Table 6.** Material properties of base metal.

| Materials | Elastic Modulus (GPa) | Yield Strength (MPa) | Strength Limit (MPa) | Density | Poisson's Ratio |
|---|---|---|---|---|---|
| PA2200 | 1.140 | 23.3 | 48.1 | 0.956 | 0.28 |
| PLA | 1.764 | 47.628 | 53.822 | 1.26 | 0.35 |
| AlSi10Mg | 25.804 | 170 | 230 | 2.7 | 0.33 |
| 20 Steel | 213 | 245 | 710.67 | 7.80 | 0.3 |

Figure 24 shows the stress–strain curves of the lattice structure with different base materials. Two lattice structures with 20 steel and Alsi10Mg as base material deform almost simultaneously and become a drum shape on each layer. However, the deformation of the other two lattice structures occurs layer by layer, and their platform areas rise gradually without fluctuation until the structure reaches densification.

The energy absorption index results of lattice structures with different base materials are shown as Table 7, and the specific energy absorption of the structure is greatly affected by base materials. The $SEA_v$ and $SEA$ values of 20 steel are about 19.5 times that of PA2200, and the platform stress is about 22.1 times that of PA2200, while the peak load is much larger than PA2200.

The specific energy absorption of the four different materials of the lattice structure has a positive correlation with the strength of the materials, and it is feasible to improve the energy absorption of the lattice structure by increasing the strength of the matrices.

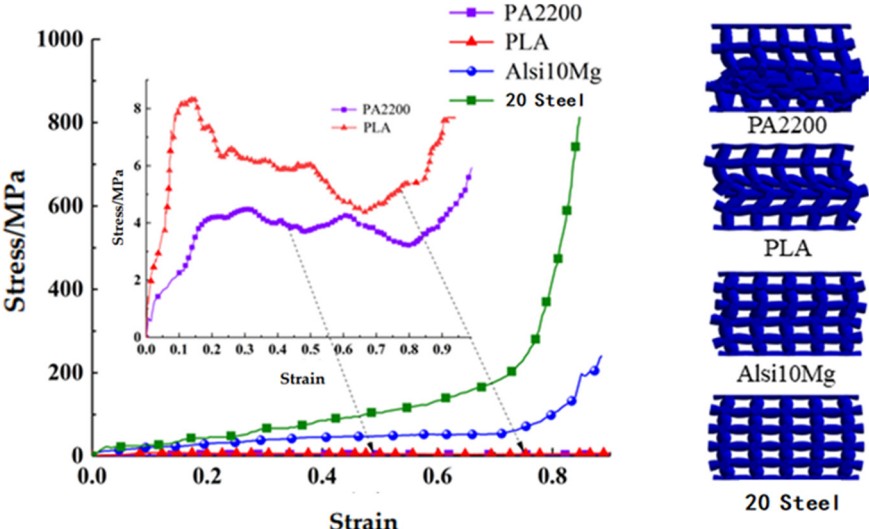

**Figure 24.** Stress–strain curves for different materials.

**Table 7.** Energy absorption index of different base metals.

| Materials | $SEA_v$/mJ·mm$^{-3}$ | $SEA$/J·g$^{-1}$ | $CLE$/% | $MIF$/KN | $\sigma_P$/MPa | $\varepsilon_d$ |
|-----------|------------------------|--------------------|---------|----------|------------------|------------------|
| PA2200 | 2.91 | 2.31 | 27.37 | 11.20 | 3.58 | 0.81 |
| PLA | 3.89 | 3.09 | 22.5 | 20.78 | 5.86 | 0.66 |
| AlSi10Mg | 29.16 | 23.14 | 35.48 | 166.3 | 38.96 | 0.75 |
| 20 Steel | 56.86 | 45.12 | 29.73 | 471.75 | 79.76 | 0.71 |

## 5. Conclusions

The lattice structures have the advantages of high strength, high stiffness, and good absorption and storage of energy. It is useful to find an optimal lattice structure with better energy absorption performance by investigating the fiber structure of loofah under the numerical analysis method. By compressing the four single-cell and VC lattice structures, the bio-90 lattice structure was found to have the best energy absorption performance. We studied the influences of the number of cells, the relative density, the impact velocity, and the material on the energy absorption performance of the bio-90 lattice structure. When the number of cells exceeds a certain value, the energy absorption performance of the overall structure will be reduced. The relative density has a significantly different influence on the energy absorption performance of lattice structures, which is the maximum when the relative density is 0.37. With the impact velocity increasing, the absorption energy rises slightly, but the energy absorption process of the structure is unstable, and the crash load efficiency (*CLE*) always decreases. It is feasible to improve the energy absorption performance of the lattice structure by replacing the material. However, if the peak load is strictly limited, it can be decreased by reducing the relative density and number of cells. The research results provide useful references for the design of efficient energy absorption structure.

**Author Contributions:** Revision, funding acquisition, and conceptualization, G.H.; methodology, G.H. and H.Y.; writing—original draft preparation, H.Y. and T.C.; writing—review and editing, G.H. and H.Y.; software, Y.N. and H.Z.; data curation, H.Z. and F.Z.; resources, H.Y. and T.C.; validation, G.H. and H.Y.; formal analysis, T.C. and H.Z. All authors have read and agreed to the published version of the manuscript.

**Funding:** This work was supported by the National Natural Science Foundation of China (Grant No. 52175223 and the Key Research and Development Program of Changzhou (Grant No. CE20225044).

**Institutional Review Board Statement:** Not applicable.

**Informed Consent Statement:** Not applicable.

**Data Availability Statement:** Not applicable.

**Conflicts of Interest:** The authors declare no conflict of interest.

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
