# Peer review of "Lattice Structure Design Method Aimed at Energy Absorption Performance Based on Bionic Design"

_machines, doi:10.3390/machines10100965_

Round 1

Reviewer 1 Report

The paper presents an interesting work on energy absorption. Some revision is required to consider it for publication.

·       What is VC? The authors should not write ACRONYMS before introducing them.

·       What is the 3D printer type? Needs to be mentioned in the abstract.

·       The authors didn’t reference any of the similar energy absorption work done by Maskery et al, Elmadih et al and/or Abueidda et al.

·       What is the accuracy of the equations used to calculate the diameter and volume? How was that checked? It is suggested that the accuracy of these methods are checked and reported.

·       The manuscript has some grammatical errors and spelling mistakes. The authors need to addres that and need to also stick to scientific writing style (for example, better to use "have not" instead of "haven't").

·       The authors need to report on the deviations expected from the manufacturing process and quantify that if possible.

·       The manuscript cites only 22 references. A more thorough literature review is required:

o   Check this paper: https://www.researchgate.net/publication/312492580_Evolution_of_full_phononic_band_gaps_in_periodic_cellular_structures

o   Check this thesis: https://nusearch.nottingham.ac.uk/primo-explore/fulldisplay?docid=44NOTUK_EPRINTS60383&context=L&vid=44NOTUK&lang=en_US&search_scope=44NOTUK_COMPLETE&adaptor=Local%20Search%20Engine&tab=44notuk_complete&query=any,contains,waiel%20elmadih&facet=rtype,exclude,reviews,lk&offset=0

Author Response

Point 1:   What is VC? The authors should not write ACRONYMS before introducing them.

Response 1: The VC is the Vertex Cube which is a model obtained by volume topology optimization of a cube used in other article to measure its performance. And in order to further explain the VC model, a simplified picture of VC is added to the article.

Point 2: What is the 3D printer type? Needs to be mentioned in the abstract.

Response 2: The 3D printer is a device of the fused deposition modeling (FDM), which adopts layer by layer printing. The 3D printer type has been further described in the abstract, the content added is “fabricated by the fused deposition modeling (FDM) 3D printer”

Point 3: The authors didn’t reference any of the similar energy absorption work done by Maskery et al, Elmadih et al and/or Abueidda et al.

Response 3: Thanks for the guidance of the reviewers, we carefully read the articles of the three authors, and cite some research contents in the article, the cited article is “Elmadih, Waiel. Additively Manufactured Lattice Structures for Vibration Attenuation. N.p., 2020. Print” and content is about the shock-absorbing performance of lattice structure.

Point 4: What is the accuracy of the equations used to calculate the diameter and volume? How was that checked? It is suggested that the accuracy of these methods are checked and reported.

Response 4: Because the overlapping part between poles and the part removed by the top of the pole that is contained in the cube are ignored, the calculation results are wrong. A new formula is obtained by theoretical derivation and polynomial fitting, the added contnent is “The calculation formula of relative density about single-cell structures is modified by theoretical derivation and polynomial fitting method”.

The primary and modified formulas are shown in the following table.

Based on the actual relative density with the difference of L obtained by the volume evaluation module of the modeling software. The deviation between the relative density calculated by the two formulas and the actual density is shown in the figure 7. The deviation of the result by the new formula is within 0.08%. And the corresponding content have been supplemented in the article.

Point 5: The manuscript has some grammatical errors and spelling mistakes. The authors need to addres that and need to also stick to scientific writing style (for example, better to use "have not" instead of "haven't").

Response 5: The grammatical and spelling problems in this article have been completely revised, for example the following three, and so on.

  • Modify” The elastic modulus and yield strength haven’t relationship with the design angle by the table 5.” into “The elastic modulus and yield strength have not relationship with the design angle by the table 5.”
  • Modity “[Ngoc San Ha, Guoxing Lu. A review of recent research on bio-inspired structures and materials for energy absorption applications[J]. Composites Part B,2020,181(18):107-122.” into “Ngoc San Ha, Guoxing Lu. A review of recent research on bio-inspired structures and materials for energy absorption applications[J]. Composites Part B,2020,181(18):107-122.”.
  • Modify ” This quasi-static compression experiment about the above five models(bio-45, bio-60, bio-75,bio-90, VC) is using the DZ-101 machine of Dazhong Instrument Co., Ltd.,” into “By the DZ-101 machine of Dazhong Instrument Co., Ltd., this quasi-static compression experiment about the above five models(bio-45, bio-60, bio-75,bio-90, VC) is experimented”.

Point 6: The authors need to report on the deviations expected from the manufacturing process and quantify that if possible.

Response 6: The precision of 3D printer is 0.05~0.2mm. In order to reduce manufacturing deviation, the printing temperature, speed, printing thickness and other parameters are all in accordance with the recommended values of the equipment. We will further study how to analyze the range of manufacturing deviation in the future. And add the content about the printing parameter setting in the article. The content added is “The precision of 3D printer is 0.05~0.2mm. In order to reduce manufacturing deviation, the printing temperature, speed, printing thickness and other parameters are all in accordance with the recommended values of the equipment.”.

Point 7: The manuscript cites only 22 references. A more thorough literature review is required: o

Check this paper: https://www.researchgate.net/publication/312492580_Evolution_of_full_phononic_band_gaps_in_periodic_cellular_structures

Check this thesis: https://nusearch.nottingham.ac.uk/primo-explore/fulldisplay?docid=44NOTUK_EPRINTS60383&context=L&vid=44NOTUK&lang=en_US&search_scope=44NOTUK_COMPLETE&adaptor=Local%20Search%20Engine&tab=44notuk_complete&query=any,contains,waiel%20elmadih&facet=rtype,exclude,reviews,lk&offset=0

Response 7: The above two papers have studied the shock-absorbing performance of lattice structures and analyzed the influence of the number of cell and structure on the performance of lattice structures, which has a lot to do with the research content of this article. These two papers have been cited, and another article has been cited:

4.Chenxi P, Phuong T, H. N-X, A.J.M. F. Mechanical performance and fatigue life prediction of lattice structures: Parametric computational approach. Composite Structures 2020; 235(C).

15.Elmadih, Waiel. Additively Manufactured Lattice Structures for Vibration Attenuation. N.p., 2020. Print.

20.Wormser, M., Warmuth, F. & Körner, C. Evolution of full phononic band gaps in periodic cellular structures. Appl. Phys. A 123, 661 (2017). https://doi.org/10.1007/s00339-017-1278-6

Reviewer 2 Report

  • The paper describes the result of an investigation aimed at obtaining lattice structures (inspired by nature) capable of absorbing energy. The manuscript is overall clear and the aim of the research is stated properly. The strength point of the paper is that experimental data are included which gives the paper a high added value with respect to literature. The main weak point of the manuscript is that some more details about the FEM tests carried out must be included in the section related to the analysis of lattice structures dynamic performances.

    Therefore a revision is recommended, keeping into account the following suggestions:

    • please define in Line 31 what BCC and further acronym stands for: a reader could not so much skilled in lattice structures to know the meaning
    • Line 96. Briefly explain what type I and II mean because the definition is provided later in the manuscript
    • Section 2: the behaviour of lattice structures depends to a large extent on the slenderness ratio of the ligaments, and the presence of rounding (or sphere used by some authors) added in the points where ligaments converge. Did you consider these effects? Could you comment on the effect of slenderness and edges on the results obtained?
    • Line 146 and section 3: Please define the type of FDM wire used for tests, and the printing direction of the structures: the FDM leads to non-isotropic materials. Did you use support material? If not explain that the structure is self-supporting.
    • Sentence lines 156 -159 I think there is a missing verb. Please check the sentence.
    • Figure 14: if it’s not by you please cite also in the figure caption
    • Section 4. Please include the FEM software you used, the finite element used to model lattice structure, the solver used, the constraints used, the total number of nodes for the structures, the computing time, eventual comparison between FEM data and experimental data, the way in which the material has been modelled (e.g. isotropic, non-isotropic) including elastic module used in simulations.

Author Response

Point 1:   please define in Line 31 what BCC and further acronym stands for: a reader could not so much skilled in lattice structures to know the meaning.

Response 1: The three models are abbreviated as the three model used for testing in reference 6, BCC is the body-centered cubic structure, SGBCC is the size-change graded body-centered cubic structure, RGBCC is the rod-diameter-change graded body-centered cubic structure. The full names of these three abbreviations are supplemented in the article.The supplementary content in the article is “Bai [6] used PA2200 as raw material, printed three experimental samples of body-centered cubic (BCC), rod-diameter-change graded body-centered cubic (RGBCC) and size-change graded body-centered cubic (SGBCC), and studied the effect of gradient direction on the structure.”

Point 2: Line 96. Briefly explain what type I and II mean because the definition is provided later in the manuscript.

Response 2: The type I represents the angle of 45 ° lattice structure, and due to the three lattice structures with angle of 60 °, 75 ° and 90 ° have uniform characteristics, they all have the middle rod. The angle is defined as the residual angle between the rod b and y axes, as shown in the figure below.

The type I and type II in the article are described in detail, “The difference between type I and type II structure is the values of angle. The type I represents the angle of 45 ° lattice structure, and due to the three lattice structures with angle of 60 °, 75 ° and 90 ° have uniform characteristics, they they all have the middle rod.”

Point 3: Section 2: the behaviour of lattice structures depends to a large extent on the slenderness ratio of the ligaments, and the presence of rounding (or sphere used by some authors) added in the points where ligaments converge. Did you consider these effects? Could you comment on the effect of slenderness and edges on the results obtained?

Response 3: The slenderness ratio is not linear with the load capacity of the structure, and there is an optimal value for different structure. Generally, the slenderness ratio is 3-6. In addition, the cross-sectional area of ligaments will be smaller in the convergence of the structure, which will lead to easy fracture in the process of tension and compression. Adding rounded corners can improve the bearing capacity of the structure. Due to the limited time, the influence of slenderness ratio and fillet on structural performance is the next research content.

Point 4: Line 146 and section 3: Please define the type of FDM wire used for tests, and the printing direction of the structures: the FDM leads to non-isotropic materials. Did you use support material? If not explain that the structure is self-supporting.

Response 4: The printing direction of the printer is bottom-up. The different printing directions of the adjacent two layers lead to structural anisotropy. The five lattice structures used in this paper all use support materials (PLA). The added content in the article is “The printing direction of the printer is bottom-up, and the different printing directions of the adjacent two layers lead to structural anisotropy. The five lattice structures used in this paper all use support materials (PLA).” .

Point 5: Sentence lines 156 -159 I think there is a missing verb. Please check the sentence.

Response 5: Sentence lines 156 - 159 have been modified, the content is “By the DZ-101 machine of Dazhong Instrument Co., Ltd., this quasi-static compression experiment about the above five models(bio-45, bio-60, bio-75,bio-90, VC) is experimented”. Other grammatical problems in the article have also been modified.

Point 6: Figure 14: if it’s not by you please cite also in the figure caption

Response 6: Thank the reviewer for your suggestions, the figure 14 is made by myself, but the figure in Figure 1 is cited and marked.

Point 7: Section 4. Please include the FEM software you used, the finite element used to model lattice structure, the solver used, the constraints used, the total number of nodes for the structures, the computing time, eventual comparison between FEM data and experimental data, the way in which the material has been modelled (e.g. isotropic, non-isotropic) including elastic module used in simulations.

Response 7:

Due to it is difficult to study the influence of the number of cells, the relative density, the impact velocity, the material on structure performance, by experiments, we have not carried out relevant research, which will be improved in the future. However, in order to effectively verify the similarity between the experimental and simulation results, we compared the stress and strain simulation results of the five models mentioned in Section 3 with the experiment results. The data shows that the deviation is small.

The specific finite element analysis content added in the article is

“The material is PLA, and the lattice structure is anisotropic, which is the same as the 3D printed lattice structure. Two steel plates are placed respectively on the upper and lower surfaces of the lattice structure. The steel plate and the lattice structure are in friction contact, with a friction coefficient of 0.15. The constraint condition is that the lower steel plate is fixed, and the rotation of the upper steel plate in X, Y and Z directions is limited. The tetrahedral element grid is adopted for grid division, with the grid size of 0.6mm and the number of elements of 106325, as shown in Fig. 14(b). The modeling and mesh generation processes of the bio-60, bio-75, bio-90, VC lattice structure are the same as the bio-45, and results of finite element analysis are shown in Fig. 15.”
